# Native Whey Induces Similar Adaptation to Strength Training as Milk, despite Higher Levels of Leucine, in Elderly Individuals

**DOI:** 10.3390/nu11092094

**Published:** 2019-09-04

**Authors:** Håvard Hamarsland, Mathias K. Johansen, Fridtjof Seeberg, Marie Brochmann, Ina Garthe, Haakon B. Benestad, Truls Raastad

**Affiliations:** 1Department of Physical Performance, Norwegian School of Sport Sciences, P.O. Box 4014 Ullevål Stadion, 0806 Oslo, Norway; 2Department of Health, Nutrition and Management, Faculty of Health Sciences, Oslo and Akershus University College of Applied Sciences, 0130 Oslo, Norway; 3Norwegian Olympic Federation, 0863 Oslo, Norway; 4Section of Anatomy, Institute of Basic Medical Sciences, University of Oslo, 0317 Oslo, Norway

**Keywords:** protein supplementation, amino acids, protein quality, resistance exercise, gerontology, mTORC-1

## Abstract

Background: Large amounts of protein (40 g) or supplementing suboptimal servings of protein with leucine are able to overcome the anabolic resistance in elderly muscle. Our aim was to compare the effects of supplementation of native whey, high in leucine, with milk on gains in muscle mass and strength during a period of strength training, in elderly individuals. Methods: In this double-blinded, randomized, controlled study, a total of 30 healthy men and women received two daily servings of 20 g of either milk protein or native whey, during an 11-week strength training intervention. Muscle strength, lean mass, m. vastus lateralis thickness, muscle fiber area, and resting and post-exercise phosphorylation of p70S6K, 4E-BP1, and eEF-2 were assessed prior to and after the intervention period. Results: Muscle mass and strength increased, by all measures applied in both groups (*p* < 0.001), with no differences between groups (*p* > 0.25). p70S6K phosphorylation increased (~1000%, *p* < 0.045) 2 h after exercise in the untrained and trained state, with no differences between supplements. Total and phosphorylated mTORC-1 decreased after training. Conclusion: Supplementation with milk or native whey during an 11-week strength training period increased muscle mass and strength similarly in healthy elderly individuals.

## 1. Introduction

Aging is associated with a loss of muscle mass and strength and if allowed to advance, this condition may proceed to sarcopenia, early loss of independent living [1], and several comorbidities [2,3]. The most promising intervention to prevent or counteract sarcopenia to date is resistance exercise in combination with appropriate nutrition with a focus on protein consumption [4]. Among the emerging causal mechanisms of sarcopenia, anabolic resistance, meaning a dampened responsiveness in skeletal muscles to the stimuli of resistance exercise and protein intake has received much attention in later years [5]. With a growing aged population, the need to develop effective interventions including exercise and nutrition is increasing. 

The effect of resistance training and protein on muscle mass is manifested through the summation of transient periods with positive net muscle protein balance leading to an accumulation of muscle mass over time [6,7]. Thus, by optimizing each period of positive net muscle protein balance, greater positive changes in muscle mass can be achieved over time. The anabolic effect of a protein is dependent on the digestion rate and amino acid kinetics [8,9,10] and amino acid composition [11,12], in particular levels of leucine [13]. Acute studies have shown promising results for high leucine or leucine-enriched proteins in terms of stimulating muscle protein synthesis [13,14,15]. 

Our lab has previously shown that native whey, which is high in leucine, increases signaling related to translation, and muscle protein synthesis (MPS) to a greater extent than milk in elderly participants after resistance exercise [16]. Effects of protein supplementation in long-term training studies are less clear (Morton 2017) [17,18,19,20]. Native whey is manufactured by the filtration of unprocessed raw milk, leaving proteins complete. Producing whey by this method removes glycomacropeptides, which are low in leucine, thus increasing the leucine concentration of native whey by about 15% compared to regular whey and 25% compared to milk [21]. 

The aim of the current study was to examine whether previously reported short-term differences in anabolic signaling and muscle protein synthesis between native whey and milk [16] translates into differences in long-term adaptations of muscle mass and strength during a 11-week strength training regime in elderly participants. We also included an acute study at the beginning and at the end of the training intervention in order to investigate potential changes in anabolic signaling between the untrained and trained state. 

## 2. Materials and Methods 

### 2.1. Participants and Ethical Approval

Thirty-eight elderly (+70 years) men and women were included in the study (Table 1). The main outcome in this study was lean mass. As no previous study had compared different protein supplements in elderly individuals at the time, we based our power calculations on a combination of protein supplementation studies in the elderly [18,19,22] and studies comparing different sources of protein supplementation in the young [23,24]. Our goal was to include 20 participants in each group to have an 80% power to detect a true difference of 1.3% with a standard deviation of 1.5%. Unfortunately, eight participants withdrew from the intervention after inclusion. One withdrew after the first acute study, after which the participant experienced dizziness. Two withdrew due to arthritic pain. Three withdrawals were due to busy time schedules. Two participants experienced health issues, not related to the study, and were excluded due to low workout attendance. Prior to participation in the study, all participants completed a medical screening. Individuals with lactose intolerance, milk allergy, allergy to local anesthesia, habit of smoking, cardiovascular disease, or any injuries to the musculoskeletal system that would impede the ability to perform strength training were excluded from this study. One participant was on anticoagulants and 6 participants took statins. The use of any dietary supplements was prohibited during the intervention. If participants were using supplements, usage was halted at least two weeks prior to pre-testing. Participants were untrained and had no prior experience with heavy resistance exercise. The study was performed in accordance with the Declaration of Helsinki and approved by the Regional Ethics Committee for Medical and Health Research of South-East Norway. A written informed consent form was signed by all participants before entering the study. The trial was registered at clinicaltrials.gov as NCT03033953. Figure 1 shows a flowchart of the study. 

### 2.2. Study Design

This study was conducted as a double-blinded, randomized, controlled trial. Randomization was conducted using an online tool (www.randomizer.org/). Participants, trainers, and researchers involved in lab analysis were blinded to group allocation until analyzes were finished. The design involved 11 weeks of strength training totaling 33 workouts. A subgroup of participants (*n* = 20) took part in an acute study during the first and last workout sessions of the training period. As a result, muscle biopsies were collected in the untrained rested, untrained acutely exercised, trained rested, and trained acutely exercised state. Participants were randomized to one of two groups, receiving either native whey or 1% fat milk. 

### 2.3. Supplements

Native whey was purchased from Lactalis^®^ (Laval, Mayenne, France) and the 1% milk was manufactured by Tine ASA (Oslo, Norway). Supplements were matched on macronutrients by adding cream (Tine ASA, Oslo, Norway) and lactose (Arla foods, Viby, Denmark) to the native whey protein. Supplements were spray dried into powder form, to be mixed with 0.4 L of water before intake. The supplements had a similar flavor and appearance. The amino acid and macronutrient content of the supplements are listed in Table 2. On training days, one serving was consumed immediately after training, and one serving was consumed in the afternoon. On days with no training, supplements were consumed in the morning and in the afternoon. Supplements were consumed every day from the first training session to the last day of testing. The decision to distribute the protein as described was made in order to optimize the anabolic effects of the supplementation over the whole day, in contrast to our previous study, aiming to maximize the anabolic effects for five hours after exercise. Adherence to the supplementation schedule was reported by the participants each training and recorded on each training day by the trainers. 

### 2.4. Daily Food Intake

Participants were asked to continue their normal dietary routine during the study. A trained dietitian conducted two 24-h dietary recall interviews with each participant at week 1, 5, and 11 during the training intervention. The dietary nutrient content was analyzed using the software Mat på Data 5.1 (Mattilsynet, Oslo, Norway, 2009). Nine participants had a protein intake below 1.0 g × kg body weight^−1^ at the first recording and were recommended to increase their protein intake and given the necessary guidance to achieve this. These participants all increased their protein intake, but four participants (two in each group) were still below 1.0 g × kg body weight^−1^ (>0.89 g × kg body weight^−1^) during the study.

### 2.5. Training Program

The training intervention consisted of a conventional whole-body resistance-training program consisting of three workouts each week. Loads ranged from 12 to 6 RM, for 1 to 3 sets and progressed from higher to lower repetition ranges during the 11 weeks of training. During the program, Mondays started at 1 to 2 sets of 12 RM for the first three weeks before adding another set to several exercises in weeks 4 to 9 and again in weeks 10 to 12. Fridays progressed from 1 to 2 sets of 8 RM in weeks 1 to 6 to 2 to 3 sets of 6 RM in weeks 7 to 12. On Mondays and Fridays, workouts were conducted with maximal training load and intensity for the given repetitions. On Wednesdays, workouts were submaximal, using 90% of the load on the previous Monday for the same amount of reps. In addition to the exercises listed under the standardized workout, participants also did weighted back-extensions and abdominal exercises at the end of each workout. Inter-set rest periods lasted for 2 to 3 min. Qualified trainers oversaw participants during all training sessions, and continuous rotation of instructors between participant groups throughout the period minimized any potential differences in coaching. 

### 2.6. Dual-Energy X-ray Absorptiometry

Body composition was assessed in the overnight fasted state by dual energy X-ray absorptiometry (Lunar iDXA GE Healtcare, Madison, Wisconsin, USA using the enCORE Software Version 14.10.022) prior to and after the intervention. No training or testing was allowed for 48 h before the scan. Whole body scans were completed in the supine position, providing values for lean tissue, fat mass, and bone mineral content. The analysis of lean mass has a coefficient of variation < 1% in our lab.

### 2.7. Ultrasonography

M. vastus lateralis thickness was assessed by ultrasound at 40% of the femur length measured from distal to proximal. At the first assessment, lines were drawn to indicate the point of measurement. After the scan, a transparent overlay was placed on the thigh. The assessments lines as well as characteristics, such as moles and scars, were copied by a permanent marker. For the post-assessments, the assessment lines were marked on the skin through punctured holes in the overlay. Previous pictures of the individual participant were displayed on a separate screen in order to relocate the probe based on characteristics in the picture, such as connective tissue and blood vessels. The average of 3 images was used for each time point. Images were analyzed with OsiriX v 5.5.1 (Pixmeo, Geneva, Switzerland).

### 2.8. Maximal Strength

One repetition maximum (1 RM) in the leg press and chest press was tested prior to and after the intervention. Familiarization to the tests was performed one week before testing. After 10 min of brisk walking on a treadmill, a range of warm-up sets were completed before both exercises with 10, 6, 3, and 1 repetitions at 50%, 70%, 80%, and 90% of the expected 1 RM, respectively. 1 RM was achieved within 2 to 5 attempts, with 2 to 3 min of rest separating attempts. Knee flexion during leg press was set to 90° and grip width in the chest press was standardized. Load increments during tests could be made by a minimum of 5 kg and 1 kg for leg press and chest press, respectively. 

Isometric unilateral voluntary maximal contractions were assessed in the seated position in a custom-made knee-extension apparatus (Gym2000, Geithus, Norway), with a 90° angle in the hip and knee joints. Chest and hip movement were restricted by a four-point harness. After three warm up sets at 25%, 50%, and 75% of the perceived maximal effort, three attempts of 5 s were performed with one minute of rest between each attempt. Force was measured with a force transducer (HMB U2AC2, Darmstadt, Germany). The best contraction was used as a result of the test. The warm-up before the MVC test consisted of 5 min on a cycle ergometer, except when tested immediately after the workout sessions for the subgroup taking part in the acute study. 

### 2.9. Functional Tests

In order to measure performance changes in daily activities, we included a timed stair climb and a timed chair rise. Familiarization to the tests were performed one week before testing. The stair climb was performed in a 2-level staircase (20 steps of length: 30 cm and height: 18 cm). Participants were instructed to walk the stairs as fast as possible without transitioning into running. No arm-swing was allowed. The stair climb was performed with three different levels of load: Body weight, wearing a 10-kg weight vest, and wearing a 10-kg weight vest while carrying two 5-kg weight discs. Participants had two attempts with each load. Performance was measured using photocells (Speedtrap 2, Brower Timing Systems, Draper, UT, USA). 

For the timed chair rise, participants were instructed to stand with their arms crossed in front of a chair (height: 47 cm), then to sit and stand five times as fast as possible. When sitting, both feet had to be lifted a few centimeters off the ground to demonstrate full transfer of weight to the chair, and when standing hips needed to be fully extended for the trial to be registered. Performance was measured using a pressure plate (Speedtrap 2, Brower Timing Systems, Draper, USA) placed on the chair and proper technique was ensured by test leaders.

### 2.10. Blood Analyses

The analysis of serum glucose, insulin, urea, and creatine kinase was done at Fürst Medical Laboratory (Oslo, Norway). Blood concentrations of amino acids were analyzed as previously described [21] with a EZfaast amino acid analysis kit (Phenomenex^®^, Torrance, CA, USA) and gas chromatography/mass spectrometry (Shimadzu QP-2010 Ultra GCMS, Shimadzu Scientific Instruments, Columbia, MD, USA).

### 2.11. Biopsy Collection and Pre-Analytical Processing

Muscle specimens were sampled from the mid portion of m. vastus lateralis by a modified Bergström technique with suction. The biopsies were grinded into a homogenate of soluble proteins (for western blot) and mounted for immunohistochemistry. Pre-analytical processing, such as freezing and homogenization, of muscle biopsies was completed as previously outlined [25]. 

### 2.12. Western Blot 

Homogenates for western blot were handled as previously described [25], quantified with ChemiDoc MP (BioRad Laboratories, Hercules, CA, USA), and analyzed with Image Lab (v5.1, BioRad Laboratories, Hercules, CA, USA). Samples were run in duplicates, and all comparisons were made within each blot. Primary and secondary antibodies are listed in Appendix A. 

### 2.13. Immunohistochemistry 

Eight-micrometer-thick cross sections were blocked for 30 min with 1% BSA (bovine serum albumin; A4503, Sigma Life Science, St Louis, MO, USA) in a 0.05% Phosphate Buffered Saline with Tween® 20 (PBS-T) solution (Cat#524650, Calbiochem, EMD Biosciences) before being incubated with antibodies against myosin heavy chain II (SC71, hybriodomabank) and dystrophin (AbCam) in a blocking solution for 2 h at room temperature. This was followed by incubation with appropriate secondary antibodies (A11005 or A11001; Life technologies, Invitrogen) for 30 min at room temperature, before being covered with a coverslip and mounted with ProLong Gold Antifade Reagent with DAPI (P36935, Invitrogen Molecular Probes, Eugene, OR, USA). Muscle sections were then and left to dry overnight at room temperature. Between stages, the sections were washed 3 × 5 min in a 0.05% PBS-T solution. Individual muscle fiber CSAs were analyzed, and measurements were calculated using TEMA software (Checkvision, Hadsund, Denmark). In total, 50 type I fibers and 50 type II fibers were analyzed on each section. 

### 2.14. Acute Strength Training Experiment

In addition to the training study, 20 participants (milk: 9 and native whey: 11) performed an acute study as their first and last workout of the training intervention. Both sessions were performed after 3 to 4 days with no strenuous physical activity. In the overnight fasted state, participants ingested a standardized breakfast (50 energy percent (E%) from carbohydrate, 8% from protein, and 42% from fat) consisting of oats (0.85 g × kg body mass^−1^), water, rapeseed oil (0.2 g × kg body mass^−1^), and 5 grams of sugar (30 kJ, 0.14 g protein, 0.36 g fat, and 0.84 g carbohydrates × kg^−1^ body mass). Each participant received an individual meal plan for the length of the acute study. The plan was based on body mass and provided participants with 30 kcal × kg body mass^−1^ and 1.3 g of protein per kg^−1^ × day^−1^. 

The workout was standardized and based on RM-load. Thus, the training load was increased from the first to the last acute bout in order to reach RM. The warm-up consisted of 10 min on a treadmill and specific warm-up sets with a submaximal weight for hammer squat, bench press, and seated rowing. The workout consisted of three sets of 10 repetitions, with a new set starting every 3 min, and included hammer squat, leg press, knee extension, chest press, seated row, one set of close-grip pull down, and two sets of shoulder press. Figure 2 outlines the timeline of the study. Supplements were ingested within 5 min after the workout. 

### 2.15. Statistics

Non-normally distributed data (D’Agostino and Pearson omnibus normality test) were log-transformed before statistical analysis. All figures show data in the original form. A two-way ANOVA with repeated measures (time × group) was applied to test group differences pre- and post-11-week training intervention and relative changes from pre to post, and between the acute experiments. Sidak and Tukey’s test was used as post-hoc tests to specify significant differences between selected groups and time points and all comparisons, respectively. As comparisons within groups for blood amino acid concentrations, glucose, insulin, urea, and creatine kinase (CK) were only made against before values and Dunnet’s test was used as a post-hoc test.

Relative changes (%) between groups from the pre- to post-training period were compared with an unpaired Student’s *t* test. Relative changes within each group were assessed with a paired Student´s *t* test. Statistical analyses were made using Prism Software (Graphpad 6, San Diego, CA, USA). All results are expressed as means ± standard deviation (SD). Statistical significance level was set at *p* ≤ 0.05.

## 3. Results

### 3.1. Participant Characteristics and Compliance

We observed no group differences in the baseline characteristics (Table 2). Participants’ attendance was 33.0 ± 0.9 in the native whey group and 32.5 ± 1.2 in the milk group. Total training load (repetitions × sets × kg × sessions) during the training period was 249,000 ± 65,000 kg for the native whey group and 252,000 ± 60,000 kg for the milk group (*p* = 0.74). Compliance to the supplementation regimen was 99 ± 1% for the native whey group and 99 ± 1% for the milk group (*p* > 0.99). Daily energy intake and protein intake (g·kg body mass^−1^·day^−1^) increased by 27% in the milk group and 39% in the native whey group. The energy percent (E%) from fat decreased in the milk group (*p* = 0.004) but not in the native whey group (*p* = 0.177). The E% from carbohydrates increased in both groups (*p* < 0.01 for both groups). No differences were observed between groups for total energy intake or macronutrients (Table 3).

### 3.2. Muscle Mass and Muscle Fiber Cross-Sectional Area

Significant muscle hypertrophy was evident for lean mass (milk: 6.3 ± 3.6%; native whey: 4.6 ± 3.4%, *p* < 0.001 for both groups), thickness of the m. vastus lateralis (milk: 7.2 ± 5.3%; native whey: 7.1 ± 5.2%, *p* < 0.001 for both groups), and type II muscle fiber cross-sectional area (milk: 34.2 ± 56.7%, *p* = 0.021; native whey: 33.8 ± 27.4%, *p* < 0.001) in both groups. No change was observed for type I muscle fibers (milk: 4.7 ± 22.0, *p* = 0.62; native whey: 2.8 ± 19.7, *p* = 0.89). There were no between-group differences for any measures of muscle growth (Table 4 and Figure 3). 

### 3.3. Muscle Strength and Performance

After 11 weeks of strength training, all participants increased 1 RM in leg press (milk: 29.1 ± 14.1; native whey: 36.0 ± 16.3%, *p* < 0.001 for both groups) and chest press (milk: 22.9 ± 18.4%; native whey: 21.5 ± 7.1%, *p* < 0.001 for both groups). We observed no differences between groups for gains in strength (Table 4). The time to completion for the stair-climb trials improved by between 3% and 4% in the milk group and 6% and 8% in the native whey group and chair rise improved by 8.1 ± 13.9% and 11.0 ± 9.2% in the milk and native whey group, respectively. We observed no differences for changes in performance between groups (Table 4).

### 3.4. Acute Experiment

The acute bout induced muscular fatigue, measured as a reduction in the quadriceps’ force-generating capacity 15 min after training (native whey: −11.8 ± 6.0% and −9.5 ± 3.0%, milk: −12.4 ± 4.3% and 10.0 ± 5.0%, in the untrained and trained state, respectively, *p* < 0.001 for all). Force-generating capacity remained significantly decreased at 24 h after the exercise bout in the untrained state with milk (−9.4 ± 6.7%, *p* < 0.001) but not with native whey (−4,2 ± 5.7%, *p* = 0.099). In the trained state, no significant differences from baseline compared to 24 h after exercise were observed with milk (−4.6%, *p* = 0.103) or native whey (−2.8%, *p* = 0.414)

There were no group differences for the recovery of muscle force-generating capacity. 

### 3.5. Blood Measures

Fasting plasma concentrations of glucose, insulin, urea, and CK did not change significantly throughout the training period. Acutely, all supplements increased plasma glucose (Figure 4A) and insulin (Figure 4B), with higher levels of glucose in the milk group at 45 (*p* = 0.002) and 60 (*p* = 0.048) min, and higher levels of insulin at 30 min (*p* = 0.033) with native whey after the training intervention. Plasma urea remained relatively stable after exercise and supplement ingestion (Figure 4C). Plasma CK increased 24 h after exercise with all supplements, with the milk group reaching higher post-exercise levels than native whey before the training intervention (Figure 4D, *p* = 0.011).

### 3.6. Amino Acid Concentrations in Blood 

Fasting concentrations of branched chain amino acids (BCAA) and some other essential amino acids (EAA, native whey: Lysine, threonine, and tryptophan; milk: Lysine, methionine, phenylalanine, and tryptophan) increased during the intervention period (data not shown). The acute increase in blood concentrations of leucine, BCAA, and EAA were higher after ingestion of native whey compared to milk at 45, 60, and 75 min after protein ingestion both in the untrained and the trained state (*p* < 0.001, Figure 5). Blood concentration levels of leucine tended to increase after milk ingestion before (45, 60, and 120 min post-exercise, *p*: 0.068–0.054), and increased after (at 120 min post-exercise, *p* = 0.004) the training intervention. The area under the curve for two hours after protein ingestion was higher in the native whey group for leucine, BCAA, and EAA compared to milk in both the untrained and trained state (*p* < 0.001). 

### 3.7. Protein Signaling

Resting levels of total (milk: −24.4 ± 21.2%, *p* = 0.009; native whey: −23.2 ± 17.7%, *p* = 0.003) and phosphorylated (milk: −49.4 ± 27.6%, *p* < 0.001; native whey: −40.6 ± 31.3%, *p* = 0.002) p70S6K decreased during the training period in both groups. Phosphorylation of p70S6K increased with both supplements in the untrained and trained state after resistance exercise and protein supplementation (Figure 6A). No group differences were observed for p70S6K phosphorylation. 

Resting total and phosphorylated levels of 4E-BP1 and eEF-2 were unchanged during the training intervention (*p* > 0.51 for all, Figure 6B,C). No acute changes in phosphorylation were observed for 4E-BP1 or eEF-2 in response to resistance exercise and protein supplementation.

### 3.8. Correlations

Changes in 1 RM leg press showed moderate correlations with changes in stair climb performance (r = −0.5 to −06, *p* < 0.005). 

## 4. Discussion

The aim of this study was to test whether supplementation with a leucine-rich native whey protein would result in improved muscular adaptations of growth and strength compared to milk supplementation during 11 weeks of resistance exercise in elderly participants. In order to investigate how signaling related to translation responded to protein intake and resistance exercise changes with training status, an acute study was performed during the first and the last bouts of the training intervention. There were three primary findings: (1) The increase in muscle mass and strength after 11 weeks of strength training was similar between groups supplemented with native whey or milk; (2) we observed no differences between supplements in the effects on phosphorylation of p70S6K, 4E-BP1, and eEF-2; and (3) total and resting phosphorylation levels of p70S6K were reduced during the training and supplementation period. 

### 4.1. Effect of Protein Type on Muscle Mass and Strength

Resistance training in combination with supplementation of milk protein or native whey was effective in terms of increasing muscle mass and strength in our elderly participants. In contrast to our hypothesis, no differences were observed between the protein supplements. Previous studies have shown that the elderly may need twice the amount of protein as the young to maximally stimulate MPS at rest (0.24 g·kg^−1^ vs. 0.40 g·kg^−1^; [26] and after moderate volume, heavy resistance exercise (40 g vs. 20 g; [27]. Furthermore, adding leucine to a suboptimal dose of protein is effective in enhancing the protein synthesis response [28], and may rescue the acute MPS in young and elderly [13,14,15]. Based on these findings, we hypothesized that, due to our supposedly suboptimal dose of protein (20 g) and the difference in leucine content between supplements, native whey would lead to superior stimulation of MPS, leading to a greater accretion of muscle mass over time than milk. 

Whey has been found to acutely stimulate MPS to a greater extent than casein in elderly that are rested [8] and have exercised [29], but not all studies report such differences [30]. A recent review suggests the anabolic resistance in elderly substantiate mainly when stimuli are suboptimal [31]. When combining heavy resistance exercise with “optimal” amounts of protein, few studies find evidence for an anabolic resistance in elderly [31]. Thus, in the present study, the combination of high-volume heavy resistance exercise and 20 g of high-quality protein may have been enough to overcome any potential anabolic resistance and made the higher leucine content in native whey redundant. 

Most previous studies investigating the effects of protein supplementation and resistance training in elderly have not reported any significant effects of protein supplementation [19,20,22,32,33,34]. On the contrary, a meta-analysis by Cermak and colleagues [17] found greater increases in lean mass and 1 RM in leg press, when resistance training was combined with protein supplementation in elderly. Similar findings were reported by a more recent meta-analysis by Morton and colleagues [35] for young but not elderly individuals. Thus, there is some discrepancy regarding what effects we could expect from our protein supplementation in elderly. 

We are not aware of previous studies comparing whey and milk/casein supplementation in combination with resistance training in elderly. In contrast to our findings, Hidayat and colleagues [36] observed a greater effect on fat-free mass in studies supplementing with whey protein, compared to casein or milk protein concentrate. Several of the studies on protein supplementation in elderly are challenged by a low number of participants, by not exclusively including elderly participants, and by a suboptimal supplementation in terms of frequency and/or amount of protein [20,22,32,33,34]. The few studies reporting additional effects of protein in elderly investigated frail elderly populations, where the potential benefits may be greater than in healthy elderly [18,37,38]. Without an “optimal” intervention it is hard to conclude on the potential effects of protein supplementation in elderly. We observed an average increase in lean mass of 2.1 kg. This is more than the previously reported 0.6 and 1.4 kg increases in lean mass [18,19,20,22,32,34,37] after protein supplementation and resistance training in elderly. This may relate to the health state of our participants, which was very good (almost no use of medication) and their high initial activity level, potentially counteracting anabolic resistance and allowing them to complete the demanding training program. By applying an RM-based program, all participants were at or close to failure on the Monday and Friday workouts, ensuring a strong growth stimulus. Unfortunately, without a placebo group, we cannot determine the separate contributions from resistance exercise and protein supplementation on strength and lean mass in our study. Nevertheless, it seems clear that as long as the dietary intake of macronutrients and protein quality is sufficient, the ability for healthy elderly to respond to strength training with large increases in muscle mass is maintained. 

In line with changes in muscle mass, we observed no differences in increases in strength or performance in the stair climb or chair rise. The 30% to 35% and 20% increase in 1 RM leg press and chest press, respectively, is similar to results from previous resistance training studies supplementing with protein in elderly [18,19,20,33,37]. Our moderate correlation between change in 1 RM leg press and change in performance indicates a transfer from strength gains to functional tasks in elderly. Importantly, we did not observe any differences between supplements for any measure of muscle mass or strength. The large variability in individual responses makes it difficult to confidently exclude a type II statistical error in the current study. 

### 4.2. Amino Acid Concentrations in Blood

Fasting levels of some individual EAA increased (5–20%) during the intervention. However, we did not see an increase in fasting levels of EAA as we observed in young individuals [39]. The previously reported association between fasting blood levels of leucine and changes in lean mass [23] was not observed in the current study. As previously reported, the acute changes in blood concentrations of leucine, BCAA, and EAA after protein ingestion were greater with native whey compared to milk [16], both in the untrained and trained state. The blood leucine concentrations after exercise and intake of native whey in the current study are comparable to values previously reported in elderly [16] but lower compared to resistance-trained young individuals ingesting 20 g of a similar native whey supplement [40]. Somewhat surprisingly, and contrary to substantial previous literature [8,29,30], we only observed tendencies towards an effect of milk ingestion on blood concentrations of amino acids within two hours after ingestion. These data are similar to what we reported in young individuals in a similar study [39], but with some more data points reaching significance. The discrepancy with previous studies is likely due to at least two factors slowing down the absorption of casein in the current study: (1) Most studies used isolated casein, whereas we utilized milk protein combined with fat and lactose to mimic the composition of skimmed milk. The addition of a milk matrix to casein have been shown to slow digestion and absorption of amino acids but not the muscle protein synthesis response [41]; and (2) although rarely stated, it is our impression that most studies use different forms of casein caseinate that are more readily absorbed than micellar casein as it exists in milk. Comparing our results to other studies using micellar casein shows good agreement [9,42,43]. It is also important to note that measuring the blood concentrations of amino acids does not allow insight into the flux of amino acids into muscle. This information would require intrinsically labelled amino acids to be ingested. 

### 4.3. Intracellular Signaling 

Despite large differences in blood concentrations of amino acids, milk and native whey supplementation resulted in comparable phosphorylation of p70S6K two hours after resistance exercise, both in the untrained and trained state. These results go against a previous study by our group, where native whey led to a greater phosphorylation of p70S6K and a concomitant greater stimulation of mixed-muscle FSR after exercise, compared to milk [16]. This could be explained by differences in the supplementation regime and timing of biopsies in our previous study, with 20 g of protein immediately after and 2 h, and biopsies at 1 and 3 h after resistance exercise. Furthermore, the difference in leucine content per serving between supplements was greater than in the current study (0.76 g vs. 0.59 g; [16]). However, blood concentrations of amino acids, including leucine, over the first 2 h after ingestion did not differ between studies. 

In contrast to Farnfield and colleagues [44], we did not show a reduced p70S6K response to protein and resistance exercise in elderly after 11 weeks of resistance training. Total and phosphorylated p70S6K in the rested state were decreased after the training intervention in both groups. Increasing age has been associated with greater basal mTORC-1 phosphorylation in both mice [45] and humans [46]. Recently, this age-related elevation of mTORC1 activity was linked to muscle fiber damage and loss in mice [47] and its inhibition by a rapalog (RAD001) showed protective effects against sarcopenia in rats [48]. Our findings suggest that resistance exercise may be capable of reducing or normalizing mTORC1 phosphorylation in aging individuals. 

### 4.4. Recovery of Force-Generating Capacity

Force-generating capacity was reduced by 5–25% after the resistance exercise protocol both in the untrained and trained state. Combined with a small increase in CK this indicates a mild to moderate muscular stress [49]. These data agrees well with a previous study, where we found no difference between milk and native whey on recovery of force-generating capacity after a “normal” bout of resistance exercise in elderly [16]. 

### 4.5. Limitations

We did not achieve the intended number of participants in the current study and the risk of committing a type 2 statistical error is present. However, we still have confidence in our conclusion as we applied several methods to measure muscle mass and strength, and all showed similar results. Differences seem to be small and although the inclusion of more participants might show significant differences, the clinical relevance of these are likely to be small. 

The post-exercise supplements were consumed immediately after exercise and had a compliance of 100%. All other supplement intakes were self-reported three times per week. Although, participants were highly motivated, we suspect an overestimation of adherence to the supplementation scheme.

By recruiting participants through the local newspaper and posters on activity centers, we likely missed the most sedate, less healthy segment of this age group. As a consequence, our results may only apply to healthy, active elderly, and not the groups that may benefit the most. 

## 5. Conclusions 

We observed no differences in muscle mass accretion or gains in strength between milk protein or native whey, when supplemented as 2 × 20 g daily servings in combination with strength training. In line with the long-term adaptations, protein supplementation after exercise increased phosphorylation of p70S6K equally with milk proteins or native whey. The resting levels of total and phosphorylated p70S6K were reduced during the training period, but the acute anabolic signaling response to resistance exercise and protein supplementation did not change from the untrained to the trained state. 

## Figures and Tables

**Figure 1 nutrients-11-02094-f001:**
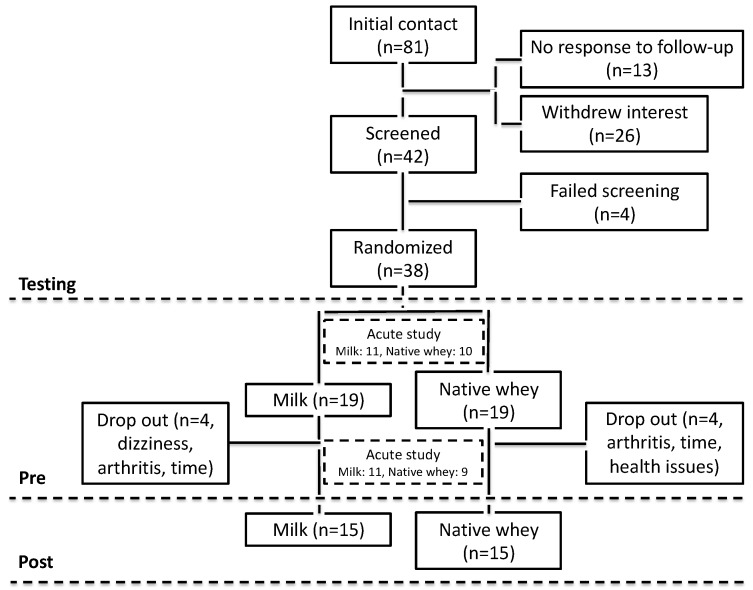
Flow chart diagram of study recruitment, enrollment, randomization follow-up, and analysis.

**Figure 2 nutrients-11-02094-f002:**
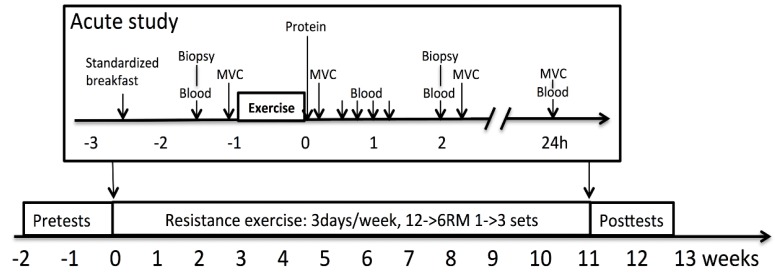
Timeline of the study.

**Figure 3 nutrients-11-02094-f003:**
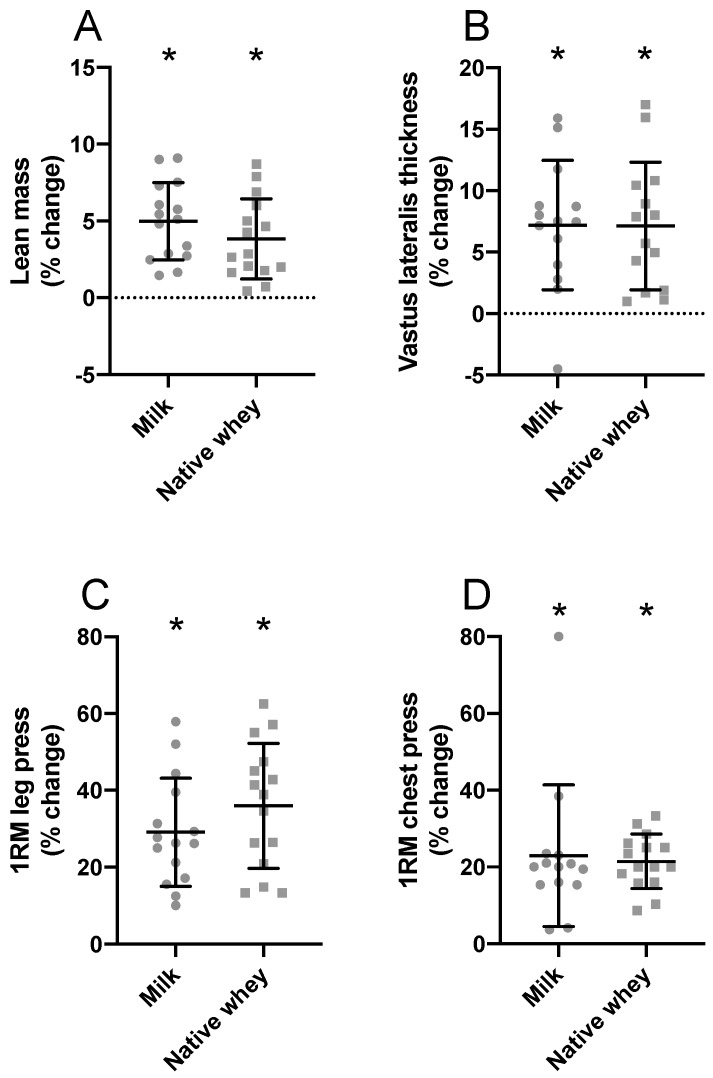
Relative changes in lean mass (**A**), m. vastus lateralis thickness (**B**), 1 RM leg press (**C**), and 1 RM bench press (**D**) after a 12-week strength training and protein supplementation period in the elderly. Values are mean ± SD. *n* = 15 and 15 in the milk group and native whey group, respectively. * different from baseline, *p* < 0.05.

**Figure 4 nutrients-11-02094-f004:**
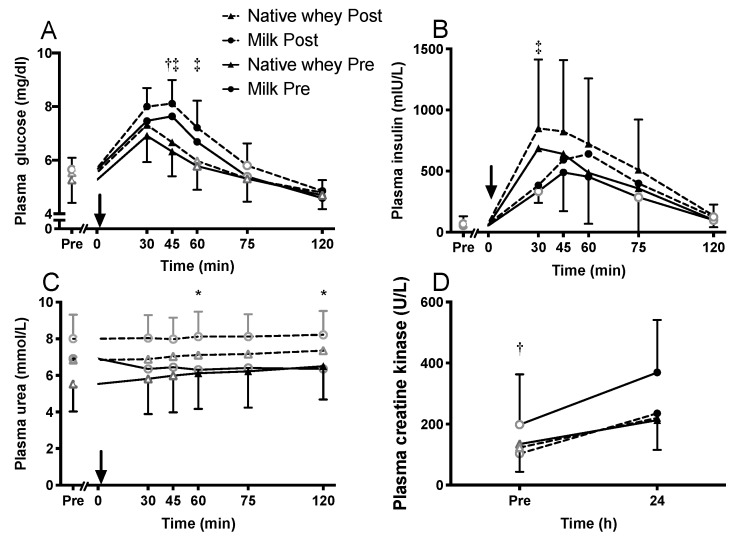
Changes in serum glucose (**A**), insulin (**B**), urea (**C**), and creatine kinase (**D**) following intake of milk or native whey immediately after a bout of resistance exercise in the elderly. Arrow indicates time point of protein supplement ingestion. Values are mean ± SD (only shown for highest and lowest values). *n* = 9 and 11 in the milk and native whey group, respectively. Data were analyzed with a two-way repeated measures ANOVA (time × supplement). Multiple comparisons tests were used as post-hoc tests to specify the significant differences between groups (Tukey) and within groups and training states (Dunett). Black data points indicate a difference form resting values; gray hollow data points indicate no significant difference from resting levels. * milk difference between pre and post, † milk and native whey different at pre, ‡ milk and native whey different at post, *p* < 0.05.

**Figure 5 nutrients-11-02094-f005:**
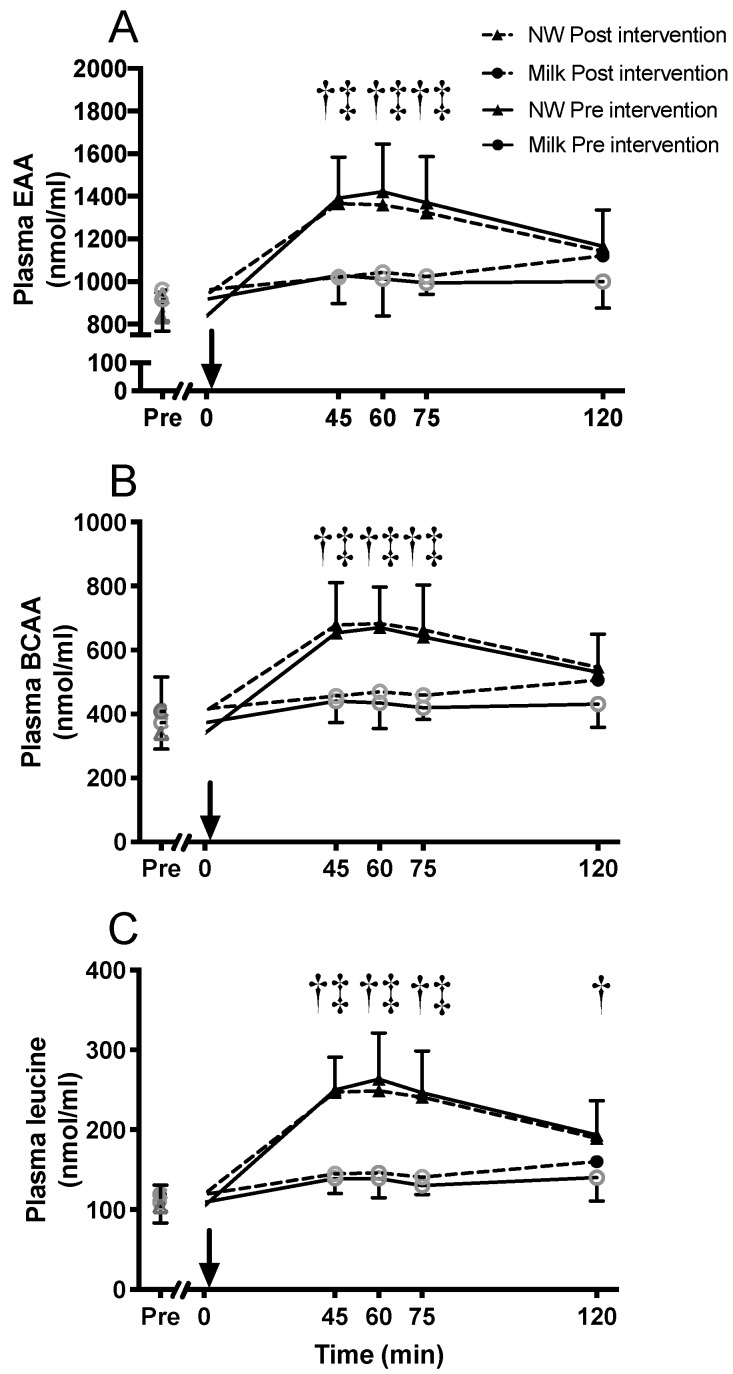
Blood concentrations of total amino acids (**A**), essential amino acids (**B**), and leucine (**C**) following intake of milk or native whey immediately after a bout of resistance exercise in the elderly. Arrow indicates time point of protein supplement ingestion. Values are mean ± SD (only shown for the highest and lowest values). *n* = 9 and 11 in the milk group and native whey group, respectively. Data were analyzed with a two-way repeated measures ANOVA (time × supplement). Multiple comparisons tests were used as post-hoc tests to specify the significant differences between groups (Tukey) and time points (Dunnet). Black data points indicate a difference form resting values; gray data points indicate no significant difference from resting levels. † milk and native whey different at pre, ‡ milk and native whey different at post *p* < 0.05.

**Figure 6 nutrients-11-02094-f006:**
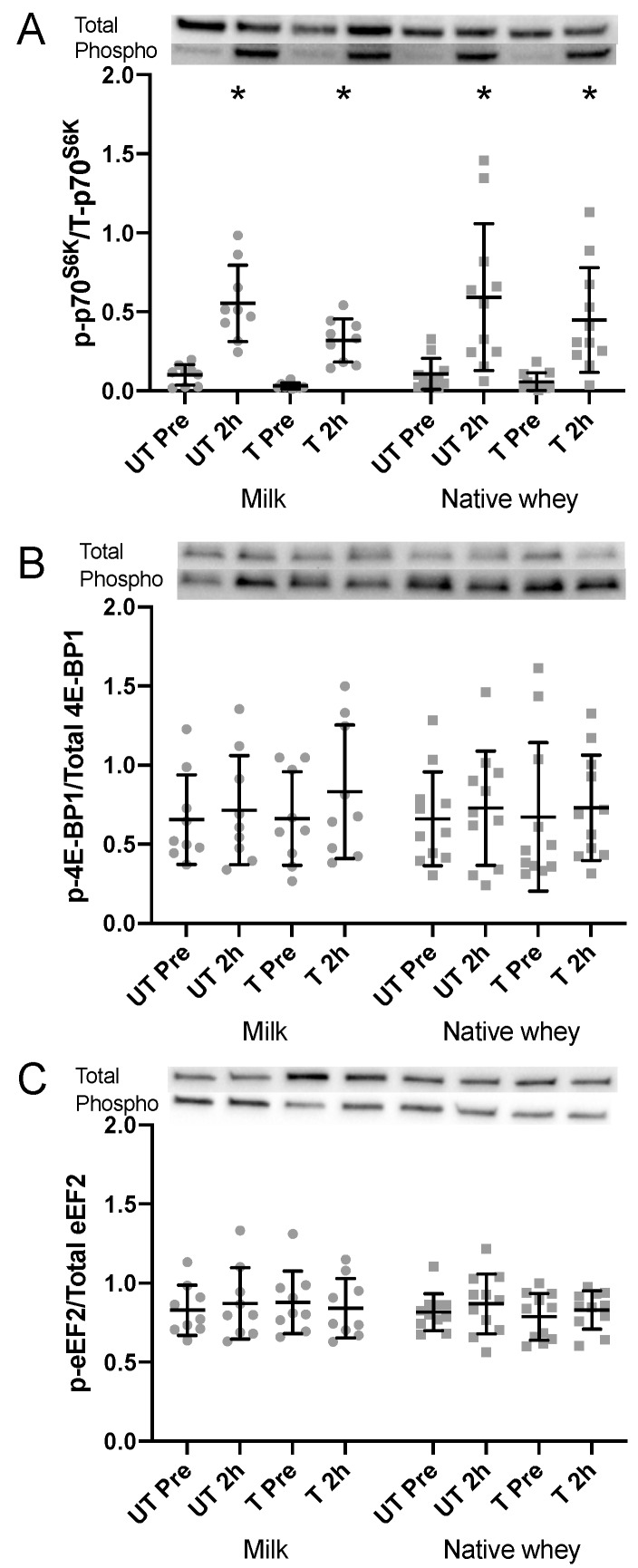
Phospho/total ratio of P70S6K (**A**), 4E-BP1 (**B**), and eEF-2 (**C**) following intake of milk and native whey immediately after a bout of resistance exercise in the elderly. UT: untrained; T: trained. Values are mean ± SD. *n* = 9 and 11 in the milk group and native whey group, respectively. Data were analyzed with a two-way repeated measures ANOVA (time × supplement). Multiple comparisons tests were used as post-hoc tests to specify the significant differences between groups (Sidak) and time points within groups (Tukey). Relative changes were analyzed with a Student’s *t* test for differences between pre and 2 h within groups (paired) and between groups (un-paired). * different from pre within group, *p* < 0.05.

**Table 1 nutrients-11-02094-t001:** Amino acid and macronutrient content of the supplements.

Amino Acids (Per Serving)	Native Whey	Milk
Alanine	0.6	1.0
Arginine	0.6	0.6
Aspartic acid	1.5	2.2
Cysteine	0.2	0.5
Phenylalanine	0.9	0.9
Glutamic acid	4.1	3.9
Glycine	0.4	0.4
Histidine	0.5	0.5
Isoleucine	1.0	1.1
Leucine	1.9	2.5
Lysine	1.6	2.1
Methionine	0.5	0.5
Proline	1.9	1.3
Serine	1.1	1.0
Threonine	0.8	1.0
Tyrosine	0.8	0.7
Valine	1.2	1.2
Tryptophan	0.2	0.4
Total protein	19.7	21.8
Fat	19.1	20.0
Carbohydrate	6.9	7.5

**Table 2 nutrients-11-02094-t002:** Participants characteristics.

	Milk	Native Whey	*p* Values for Group Differences
N (♂/♀)	15 (9/6)	15 (9/6)	
Age (years)	74.3 ± 3.6	72.9 ± 1.8	0.18
Body mass (kg)	74.6 ± 14.0	78.3 ± 16.2	0.81
Fat mass (kg)	22.1 ± 7.0	23.9 ± 8.9	0.55
Lean body mass (kg)	49.8 ± 9.2	49.5 ± 10.9	0.95
Body fat (%)	30.5 ± 5.5	32.0 ± 9.1	0.60
VL thickness (cm)	2.1 ± 0.4	2.3 ± 0.5	0.20
Leg press 1 RM (kg)	176 ± 55	158 ± 50	0.36
Bench press 1 RM (kg)	46.8 ± 20	42.8 ± 17	0.56

**Table 3 nutrients-11-02094-t003:** Daily intakes of energy and macronutrients before and during the training intervention. Values are averaged from two 24-h recall interviews. * signifies change from pre. Significance level at *p* ≤ 0.05.

	Milk		Native Whey	
	Baseline	Intervention	Baseline	Intervention
Energy (KJ)	7800 ± 2100	9400 ± 1500 *	7400 ± 1900	9800 ± 1500 *
Protein g·kg body mass^−1^	1.0 ± 0.3	1.3 ± 0.2 *	1.1 ± 0.3	1.3 ± 0.4 *
Protein (E%)	17 ± 4	17 ± 3	19 ± 3	17 ± 2
Carbohydrate (E%)	39 ± 5	46 ± 3 *	39 ± 6	43 ± 5 *
Fat (E%)	44 ± 8	37 ± 4 *	42 ± 7	40 ± 6

**Table 4 nutrients-11-02094-t004:** Changes in regional muscle mass, m. vastus lateralis thickness, muscle fiber area, and myonuclei in elderly men and women receiving milk or native whey supplementation for 11 weeks combined with strength training.

	Milk			Native Whey			*p* Values for Group Difference (% Change)
	Pre	Post	% Change	Pre	Post	% Change	
Body mass (kg)	74.6 ± 14.0	77.2 ± 14.0	3.6 ± 2.2 *	75.9 ± 16.1	78.3 ± 16.2	3.3 ± 1.6 *	0.618
Fat mass (kg)	22.1 ± 7.0	22.4 ± 6.8	1.7 ±5.3	23.9 ± 8.9	24.4 ± 8.9	2.4 ± 4.9	0.721
Leg lean mass (kg)	17.0 ± 3.8	18.0 ± 3.8	6.3 ± 3.6 *	17.3 ± 4.7	18.0 ± 4.5	4.6 ± 3.4 *	0.199
Arm lean mass (kg)	5.6 ± 1.5	6.0 ± 1.6	6.4 ± 3.6 *	5.3 ± 1.5	5.6 ± 1.5	5.2 ± 3.9 *	0.377
Trunk lean mass (kg)	24.0 ± 3.9	24.8 ± 3.7 *	3.4 ± 3.6 *	23.7 ± 4.8	24.4 ± 5.0 *	3.0 ± 3.1 *	0.791
VL thickness (cm)	2.06 ± 0.36	2.21 ± 0.39 *	7.2 ± 5.3 *	2.28 ± 0.51	2.43 ± 0.50 *	7.1 ± 5.2 *	0.971
Stair climb							
BW (s)	7.48 ± 1.0	7.18 ± 0.97	−3.8 ± 5.4 *	7.54 ± 0.94	7.03 ± 0.93	−6.56 ± 7.1 *	0.257
10 kg (s)	7.35 ± 1.0	7.21 ± 1.07	−3.3 ± 5.0 *	7.62 ± 1.34	7.09 ± 1.07	−6.29 ± 8.1 *	0.267
20 kg (s)	7.91 ± 1.53	7.58 ± 1.50	−3.8 ± 8.4 *	8.07 ± 1.73	7.41 ± 1.26	−7.26 ± 8.57 *	0.293
Sit to stand (s)	7.0 ± 2.1	6.29 ± 1.3	−8.1 ± 13.9 *	6.70 ± 1.20	5.92 ± 0.97	−11.0 ± 9.2 *	0.504
MFA type I (μm^2^)	4519 ± 1030	4667 ± 1187	4.7 ± 22.0	4897 ± 919	4933 ± 785	2.8 ± 19.7	0.823
MFA type II (μm^2^)	3862 ± 1984	4502 ±1397	34.2 ± 56.7 *	3740 ± 1608	4692 ± 1441 *	33.8 ± 27.4 *	0.935

Pre and post values are means ± SD. Changes are percent ± SD; * different from baseline, *p*-value (*p* < 0.05). MFA, muscle fiber area 1 RM, one repetition maximum; VL, vastus lateralis.

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
