# Peer review of "Native Whey Induces Similar Adaptation to Strength Training as Milk, despite Higher Levels of Leucine, in Elderly Individuals"

_nutrients, 2019, doi:10.3390/nu11092094_

Round 1

Reviewer 1 Report

Hamarsland H et al., compared the effects of native whey and milk protein on muscle strength and intracellular signaling induced by weight training. 

Major points.

1) In line 63, the authors should describe how they decided the required number of participants.  This was an important point because the author did not observe any differences between native whey and milk protein group.

2) The authors should add at least one representative image of immunohistochemical analysis and Western blot analysis for methodological validation.

3) In line 83, the authors should describe who was blinded in this study.  Furthermore, the authors should describe how they perform randomization.

4) In line 189, the list of used antibodies was not included.  The authors should describe the information about the used antibodies with their cat. number.  Furthermore, the authors should describe which phosphorylation site was analyzed.

Minor points.

5) In line 37, need an explanation for "anabolic resistance" with reference.

6) In line 49, need an explanation for MPS.

7) In line 234, "table 1" might be "table 2".

Author Response

Major points.

1) In line 63, the authors should describe how they decided the required number of participants.  This was an important point because the author did not observe any differences between native whey and milk protein group.

Authors: Thank you for pointing out this important point. We have included the basis for our power calculations and added a part on our statistical power in the limitation section.

2) The authors should add at least one representative image of immunohistochemical analysis and Western blot analysis for methodological validation.

Authors: Representative blots have been added to figure 6.

3) In line 83, the authors should describe who was blinded in this study.  Furthermore, the authors should describe how they perform randomization.

Authors: These are important points to clarify, thank you for pointing this out. This has now been added to the methods section (page 3).

4) In line 189, the list of used antibodies was not included.  The authors should describe the information about the used antibodies with their cat. number.  Furthermore, the authors should describe which phosphorylation site was analyzed.

Authors: Thank you for pointing this out. The table (S2) is now included. Phosphorylation sites are described in the table.

Minor points.

5) In line 37, need an explanation for "anabolic resistance" with reference.

Authors: The sentence has been made more explanatory and a reference has been added.

6) In line 49, need an explanation for MPS.

Authors: Muscle protein synthesis has been added

7) In line 234, "table 1" might be "table 2".

Authors: Yes, the table number is now corrected  

Authors: Additionally, we corrected a number error in figure 1 and erroneous values for muscle thickness of vastus lateralis in table 4 and figure 3B.

Authors: We would like to thank this reviewer for a fair and thorough critique. We believe that the manuscript is significantly improved based on your comments and suggestions.

Reviewer 2 Report

Comparisons of myotropic effects between whey proitein vs casein have been extensively investigated by vast numbers of researchers, and these authors were working on the case of the long-term training. This kind of cases have less been examined compared with the acute studies, which is important. However, I think there are many flaws in terms of study design, presentation of the results, and conclusion drawn from the data obtained, and they should be addressed extensively.

Major issues

First and foremost, the 3 conclusions drawn here are quite disappointing. The first finding is that muscle mass and strength were similar in the native whey and milk groups. As the authors admit in the manuscript, this study does not have a placebo group, and it is hard to conclude that the effects of these materials were similar to each other. The second one is that "no difference was seen in the phosphorylation signaling of proteins tested between the groups. This cannot be viewed as a study finding, or rather should be viewed as a defect of the experimental design. When considering the facts that muscle masses were increased and muscle strength and performance were improved in both of the groups, it is quite likely that phosphorylation signaling was modulated by the supplement and/or training. The third finding is that total and phosphorylated p70S6K were reduced during the training and supplementation period. However, this observation was only supported by data not shown, and I feel very wired about the authors indicating this observation as one of the major finding. I consider there was practically no scientific finding in this study. 

According to the Results section and Figure Legends, the authors mentioned statistical significance of various data, but I cannot see the corresponding symbols in the Figures. Thus, descriptions and Figures are quite discordant regarding statistical significance. Furthermore, I cannot understand what comparisons were made in some of the data; e.g., Line 303-305, I am wondering whether AA level were compared as AUC or the levels at specific time points.

I cannot find antibodies used for western blotting in Table S1.

Minor 

The name of the manufacturers should be followed by city, (state), and the country, when it appeared for the first time. Please and correct and pay attention to the consistency.

It seems that the authors used Endnote or something like that. Some of the references numbers are garbled. Please correct.

Some of the abbreviations were not defined when they first appeared.

Author Response

Comparisons of myotropic effects between whey proitein vs casein have been extensively investigated by vast numbers of researchers, and these authors were working on the case of the long-term training. This kind of cases have less been examined compared with the acute studies, which is important. However, I think there are many flaws in terms of study design, presentation of the results, and conclusion drawn from the data obtained, and they should be addressed extensively.

Major issues

First and foremost, the 3 conclusions drawn here are quite disappointing. The first finding is that muscle mass and strength were similar in the native whey and milk groups. As the authors admit in the manuscript, this study does not have a placebo group, and it is hard to conclude that the effects of these materials were similar to each other.

Authors: We agree that inclusion of a control group would have improved the design of the study. However, the main goal of the study was to compare the two different protein supplements. In order to have a stronger comparison between these a control group was omitted. At the time of the design of the study it seemed that protein supplementation had a beneficial effect on adaptations to strength training (https://www.ncbi.nlm.nih.gov/pubmed/23134885). Thus, a focus on optimization of protein supplementation was warranted, comparing different supplements (https://www.ncbi.nlm.nih.gov/pubmed/9838975, https://www.ncbi.nlm.nih.gov/pubmed/11818183). We would argue that the comparison (unpaired t-test) of changes in measures of muscle mass and strength between groups are valid even in the absence of a control group. This being said there is of course a possibility of a difference between groups, smaller than what could be measured in the current study. If we assume the observed differences between groups in the current study to be close to correct, we would need more than 250 participants in each group to be able to detect a statistically significant difference. The argument for no clinically relevant difference is further substantiated by several measures of muscle mass and strength pointing in the same direction.

The second one is that "no difference was seen in the phosphorylation signaling of proteins tested between the groups. This cannot be viewed as a study finding, or rather should be viewed as a defect of the experimental design. When considering the facts that muscle masses were increased and muscle strength and performance were improved in both of the groups, it is quite likely that phosphorylation signaling was modulated by the supplement and/or training.

Authors: We see our wording in the result section was somewhat unclear and we have made some adjustments to clarify this. Importantly, there was a clear expected effect of resistance exercise and protein supplementation on the acute phosphorylation of P70S6K in both groups, but no difference between groups. This agrees well with the similar growth and strength responses between groups over time. Although, the association between a snap shot of acute signaling and long-term outcomes are questionable. Regarding the other phosphoproteins we did not see an effect of resistance exercise and supplementation. The response of 4E-BP1 and eEF-2 to the stimuli of resistance exercise and supplementation are less clear in the current literature. We expected to see an increase in 4E-BP1 phosphorylation as it has been shown by other studies (https://www.ncbi.nlm.nih.gov/pubmed/22148961 https://www.ncbi.nlm.nih.gov/pubmed/22451437;  https://www.ncbi.nlm.nih.gov/pubmed/24384983; https://www.ncbi.nlm.nih.gov/pubmed/25644339). However, not all studies find this acute phosphorylation (https://www.ncbi.nlm.nih.gov/pubmed/17634259;  https://www.ncbi.nlm.nih.gov/pubmed/21045172; https://www.ncbi.nlm.nih.gov/pubmed/21795443; https://www.ncbi.nlm.nih.gov/pubmed/23343671;  https://www.ncbi.nlm.nih.gov/pubmed/23343676;  https://www.ncbi.nlm.nih.gov/pubmed/23459753;  https://www.ncbi.nlm.nih.gov/pubmed/24586775). In theory, P70S6K and 4E-BP1 should be activated in a coordinated manner, but studies have shown 4E-BP1 to be unaffected by rapamycin treatment in contrast to P70S6K (https://www.ncbi.nlm.nih.gov/pubmed/18955708; https://www.ncbi.nlm.nih.gov/pubmed/19150980). Further, 4E-BP1 seems to be dephosphorylated during exercise and may stay suppressed for some time after exercise (https://www.ncbi.nlm.nih.gov/pubmed/18056791; https://www.ncbi.nlm.nih.gov/pubmed/21045172; https://www.ncbi.nlm.nih.gov/pubmed/21795443; https://www.ncbi.nlm.nih.gov/pubmed/27053525) and the timepoint of biopsy thus may affect the result of the study. Lastly, it is possible that 4E-BP1 was already somewhat elevated by the standardized breakfast. As for eEF-2 there are studies reporting either no change (https://www.ncbi.nlm.nih.gov/pubmed/19299575; https://www.ncbi.nlm.nih.gov/pubmed/20874802; https://www.ncbi.nlm.nih.gov/pubmed/23459753; https://www.ncbi.nlm.nih.gov/pubmed/24284442; https://www.ncbi.nlm.nih.gov/pubmed/25644339) or a reduced phosphorylation (https://www.ncbi.nlm.nih.gov/pubmed/19150856; https://www.ncbi.nlm.nih.gov/pubmed/20070283; https://www.ncbi.nlm.nih.gov/pubmed/20844186; https://www.ncbi.nlm.nih.gov/pubmed/24870574; https://www.ncbi.nlm.nih.gov/pubmed/27053525; https://www.ncbi.nlm.nih.gov/pubmed/27780819) with resistance exercise and protein supplementation. We have not been able to see a pattern to explain these discrepancies.

We therefore consider our results to be within what should be expected based on the literature.

The third finding is that total and phosphorylated p70S6K were reduced during the training and supplementation period. However, this observation was only supported by data not shown, and I feel very wired about the authors indicating this observation as one of the major finding. I consider there was practically no scientific finding in this study. 

Authors: We agree that the “data not shown” statement is misplaced and will remove it. The data are reported in the result section, but not in a figure. As we already have quite a few figures in the article we reported this finding only in text. A figure can be included if this is the wish of the reviewer.

According to the Results section and Figure Legends, the authors mentioned statistical significance of various data, but I cannot see the corresponding symbols in the Figures. Thus, descriptions and Figures are quite discordant regarding statistical significance.

Authors: Thank you for pointing out this disagreement. We have corrected the mismatch between figures and figure texts and made the symbols bigger and made the gray symbols hollow in order to better differ them from the black symbols.

Furthermore, I cannot understand what comparisons were made in some of the data; e.g., Line 303-305, I am wondering whether AA level were compared as AUC or the levels at specific time points.

Authors: Thank you for pointing out this unclarity. We have now specified when we compare concentrations on specific timepoints and when we compare area under the curve.

I cannot find antibodies used for western blotting in Table S1.

Authors: Table S1 is now included at the end of the manuscript

Minor 

The name of the manufacturers should be followed by city, (state), and the country, when it appeared for the first time. Please and correct and pay attention to the consistency.

Authors: Thank you, we have now corrected this inconsistency.

It seems that the authors used Endnote or something like that. Some of the references numbers are garbled. Please correct.

Authors: The references look normal in our version. They are probably garbled by the online conversion to the file made for the reviewers. Unfortunately, we do not know if we or how we can change this.

Some of the abbreviations were not defined when they first appeared.

Authors: We have now defined muscle protein synthesis in the introduction.

Authors: Additionally, we corrected a number error in figure 1 and erroneous values for muscle thickness of vastus lateralis in table 4 and figure 3B.

Authors: We would like to thank this reviewer for a thorough critique. We believe that the manuscript is significantly improved based on your comments and suggestions.

Round 2

Reviewer 1 Report

The authors fully answered my questions.

Author Response

Thank you for a fair critique. Your comments improved the manuscript.

Reviewer 2 Report

The authors replied to my comments extensively with my citations. Still, 1) I cannot understand the rationale for the conclusion that the effects of native whey and milk were similar to each other. 2) In case the author regards that the results related to phosphorylation shown here are "within what should be expected based on the literature", I cannot see anything that is scientifically sounding in this manuscript. 

If the main objective of this study was on the differences between the supplements, the authors should re-write the manuscript extensively, focusing on the differences and discuss it. Moreover, as placebo group was not placed in this study, the authors should cite previous studies that had placebo groups, extrapolate the observations there to this study, and discuss it.

I suggest the manuscript should be written more concisely but carefully.

Author Response

The authors replied to my comments extensively with my citations. Still,

- 1) I cannot understand the rationale for the conclusion that the effects of native whey and milk were similar to each other.

Authors: All our presented data, taken together, indicate that there were no significantly different responses to the two fortification procedures. What are the specific arguments against this interpretation?

- 2) In case the author regards that the results related to phosphorylation shown here are "within what should be expected based on the literature", I cannot see anything that is scientifically sounding in this manuscript. 

Authors: This conclusion lacks a specific justification. It is not possible to respond to a referee's admittance of lack of comprehension.

- If the main objective of this study was on the differences between the supplements, the authors should re-write the manuscript extensively, focusing on the differences and discuss it.

Authors: As stated in the last paragraph of the introduction the project was designed to examine the possible difference between the supplements - which has clearly been understood be the first reviewer - so this request to extensively rewrite the manuscript is impossible to comprehend by us.

- Moreover, as placebo group was not placed in this study, the authors should cite previous studies that had placebo groups, extrapolate the observations there to this study, and discuss it.

Authors: We have responded to this lack of placebo group before. The aim of the study was to compare the two different supplements. Although we agree a control group would strengthen the study, this comparison does not necessitate a control group. We have as an alternative control (to the non-existing placebo group) given a short discussion of the use of the original measurements before starting the specific fortifications, combined with earlier results that no significant changes of such basic variables take place during a study period like the present one in the paragraph starting at line 386.